# CONTINUOUS ADAPTATION IN MULTI-AGENT COMPETITIVE ENVIRONMENTS

## ABSTRACT

In a multi-agent competitive environment, we would expect an agent who can quickly adapt to environmental changes may have a higher probability to survive and beat other agents. In this paper, to discuss whether the adaptation capability can help a learning agent to improve its competitiveness in a multi-agent environment, we construct a simplified baseball game scenario to develop and evaluate the adaptation capability of learning agents. Our baseball game scenario is modeled as a two-player zero-sum stochastic game with only the final reward. We propose a modified Deep CFR algorithm to learn the strategies of agents in a half-inning game. We also propose a strategy adaptation mechanism that continuously updates strategies based on the anticipation of the opponent's strategy in the inference time. We form several teams, with different teams adopting the same adaptation mechanism but different initial strategies, trying to analyze (1) whether an adaptation mechanism can help in increasing the winning percentage and (2) what kind of initial strategies can help a team to get a higher winning percentage. The experimental results show that the winning percentage can be increased for the team with an initial strategy learned from the modified Deep CFR algorithm. Nevertheless, those teams with deterministic initial strategies actually become less competitive.

## 1 INTRODUCTION

Reinforcement learning has been successfully employed to solve various kinds of decision making problems, such as game playing (Silver et al., 2017; 2018), robotics (Levine et al., 2016), and operation management (Han et al., 2017). An RL method typically finds the optimal strategy through the interactions with a stationary environment, which is usually modeled as an MDP process. Nevertheless, in the real world, there could be multiple learning agents in the same scenario. The interactions among these learning agents may make the environment no longer stationary from the standpoint of each individual agent. Besides, if these agents are in a competitive relationship, we would expect an agent who can quickly adapt to environmental changes may have a higher probability to beat other agents. To discuss whether the adaptation capability can help a learning agent to improve its competitiveness in a multi-agent environment, we choose a simplified baseball game as the scenario to develop and evaluate the adaptation capability of learning agents.

A lot of games, like Chess (Silver et al., 2018), Go (Silver et al., 2017) and Atari games (Bellemare et al., 2013; Mnih et al., 2015; Wang et al., 2016), only try to find the optimal action at each state. In comparison, some complicated games, like baseball and basketball games, need to take into account not only the current status but also the opponent's possible strategies. Moreover, the opponent's strategy is typically time-varying. Hence, players should not only determine what to do under different situations, but also need to dynamically adjust their strategies based on the observations from the opponent's past actions. In a typical baseball game, for example, two competitive teams play against each other based on their pre-determined initial strategies at the beginning of the game. As the game proceeds, the teams continuously update their strategies based on the actions of their opponents, trying to win the game. However, baseball games are inherently highly uncertain while the number of interactions between the pitcher and the batters is rather limited. It is very tricky for the teams to properly adjust their strategies based on a small number of interactions.

In this work, to discuss the adaptation issue in a multi-agent competitive environment, we intentionally construct a simplified baseball game scenario based on the MLB data on Statcast Search (MLB). We make the following assumptions to simplify the problem. (The simple baseball rule is shown on Appendix D.)

1. We only focus on batting and pitching in our game scenario and treat fielding as the environment which is based on the MLB data on Statcast Search.
2. We assume there are only one pitcher & batter in each team, rather than the typical 9-batter case.
3. We assume the pitcher and the batter have the same performance in pitching and batting across different teams. The only difference is their strategies. Besides, they always perform normally and we do not consider the case of abnormal performance in all games.
4. Both the pitcher and the batter have only five possible actions, as explained in the next section.

Based on the above simplifications, we manually form 13 teams, with different teams adopting different playing strategies. We also propose a modified version of the Deep CFR (Counterfactual Regret Minimization) (Brown et al., 2019) algorithm to learn the strategies of the batter and pitcher. In total, we form 14 teams to analyze their adaptation capabilities.

In our simulation, each of these 14 teams plays the best-of-three games against every other team for many series. At the beginning of each series, each team plays based on its initial strategies. As the game proceeds, each team follows the same strategy adaptation mechanism based on the observed actions of its opponent. In our simulation, there are only three games at most for each pair of teams to adjust their strategies. We then analyze the following two main issues about strategy adaptation.

**1. With a small number of observations (three games at most for each pair of teams), can the adaptation mechanism help in increasing the winning percentage?**
**2. If two competitive teams adopt the same adaptation mechanism during the game, what kind of initial strategies can help a team to get a higher winning percentage?**

## 2 BACKGROUNDS

### 2.1 SCENARIO DESCRIPTION

We first explain how we define a simplified baseball game for the analysis of strategy adaptation in a multi-agent scenario. Even though all the following discussions are based on the baseball game scenario, the deduced conclusions can be useful for similar multi-agent problems as well.

In every play of the baseball game, the pitcher aims at an expected target location and selects a pitch type. On the other hand, the batter looks for specific types of pitches appearing in some preferred attack zones. Both the pitcher and the batter select their actions for each pitch and, depending on the result of their actions, the game proceeds to the next state. Specifically, we treat our baseball game scenario as a two-player zero-sum stochastic game (multi-agent MDP) (Bowling, 2000) with only the final reward (win or lose).

We first mention two examples to explain players' different strategies under different situations.
Situation 1: 0 out, bases empty, 3 ball, 1 strike
Under this situation, the batter tends to be more selective or may even wait for a walk. On the other side, the pitcher tends to pitch a fastball to the middle of the strike zone to avoid a walk. However, if the pitcher has pitched to the middle too often, the batter may look for pitches in this zone to hit. This will increase the probability of solid contact.
Situation 2: 1 out, bases full, 3 ball, 1 strike
Under this situation, the batter tends to be more aggressive since it is a good chance to score. On the other hand, the pitcher might take advantage of the batter's aggressiveness and pitch to the corner to fool the batter. However, if the pitcher has pitched to the corner too often, the batter may shrink his attack zone to avoid being fooled.

In Figure 1, we illustrate the flow of our baseball game. The pitcher and the batter are the two learning agents to be trained to play against each other. Here, the Pitching Control represents the pitcher's ability in ball control. As the pitcher aims to pitch at a target location, we model his Pitching Control as a Gaussian distribution centered at the target location with a manually pre-determined variance. The actual pitched location is then sampled from this distribution.

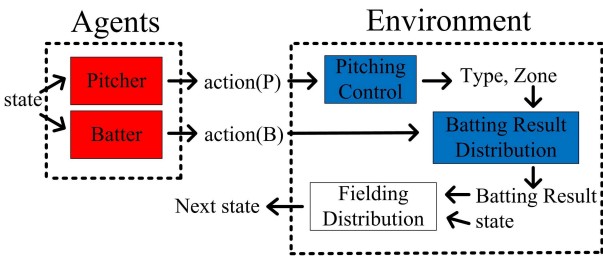

Figure 1: Flow of baseball game scenario.

On the other hand, the Batting Result Distribution $P(results|zones, type, a_b)$ models the probability of the possible batting results. Here, we classify all the possible batting $results$ into 32 categories, listed in Appendix C, with respect to different $zones$ (represented by different numbers in Figure 2), pitch $type$, and the batter's action $a_b$. To model this Batting Result Distribution, we first obtain the distribution $P(results|zones, type)$ based on the MLB data on Statcast Search. After that, we heuristically convert the distribution $P(results|zones, type)$ into the distribution $P(results|zones, type, a_b)$ for every possible action $a_b$ according to the athletic ability of the batter. Since the average MLB fastball travels at 95 mph, reaching home plate in just 0.4 second, it is very difficult for human beings to react within such a short period. Hence, the batter typically focus on a specific zone and a specific pitch type to swing in order to increase the probability of solid contact. Generally, the smaller the batter sits on the hitting zone, the more chance he can make a solid contact if the ball goes into the pre-selected zone. In Figure 1, the fielding distribution models the probability of the next state when a batting result occurs at a certain state. Again, this distribution is modeled based on the MLB data on Statcast Search.

In our baseball game, we denote each state $s$ by a 6-tuple vector: (inning, runs, outs, runners on, strike, ball). To simplify the game, both the pitcher and the batter have only five possible actions, as listed in Table 1. These actions are defined based on the pitcher's Pitch Location and the batter's Attack Zone, as illustrated in Figure 2. Here, we assume the pitcher only aims at three possible spots (the green dots) on pitcher's Pitch Location and only has two types of pitch, fastball or curveball. Due to the uncertainty in pitch control, the pitched ball actually spreads around these three spots. On the other hand, based on the pitch type and the location of the pitched ball, the batter has five possible actions.

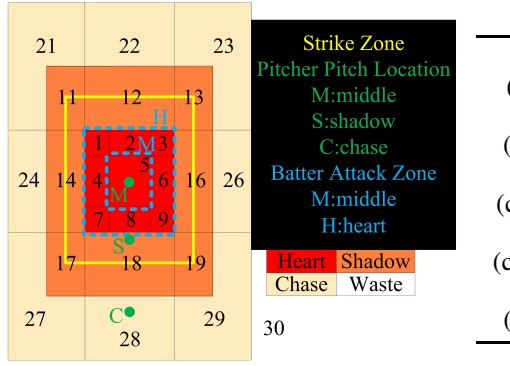

Figure 2: Strike Zone.

| Pitcher's action | Batter's action |
|---|---|
| **1. fastM** | **1. Wait** |
| (fastball in middle) | (do not swing) |
| **2. fastS** | **2. fastM** |
| (fastball in shadow) | (sit on fast ball in middle) |
| **3. curveM** | **3. fastH** |
| (curveball in middle) | (sit on fastball in heart) |
| **4. curveS** | **4. curveH** |
| (curveball in shadow) | (sit on curveball in heart) |
| **5. curveC** | **5. Any** |
| (curveball in chase) | (hit any pitch in strike) |

Table 1: Possible actions.

## 2.2 RELATED WORK

Humans are able to learn new skills quickly based on the past experience. It is necessary for artificial agents to do the same. Meta-learning, also known as "learning to learn", aims to train a model on a variety of learning tasks so that it can solve new learning tasks or adapt to the new environments rapidly with minimal training samples. Meta-learning has been used to learn high-level information of a model such as learning optimizers for deep networks (Ravi & Larochelle, 2017;

Andrychowicz et al., 2016; Li & Malik, 2016), learning task embeddings (Vinyals et al., 2016; Snell et al., 2017), and learning to learn implicitly via RL (Duan et al., 2016; Wang et al., 2016). Especially, model-agnostic meta-learning (MAML) (Finn et al., 2017b) aims to find a set of highly adaptable parameters that can be quickly adapted to the new task. The goal of quick adaptation to the new environments for meta-learning is similar to our work, but meta-learning discusses only the case of single learning agent, instead of the two-agent case. On the other hand, Maruan et al. (Al-Shedivat et al., 2018) construct an adversarial multi-agent environment, RoboSumo, allowing the agent to continuously adapt to the changes of the opponent's strategies. However, their approach considers only one-way adaptation and the opponent is not allowed to anticipate the learning agent's strategy. This is different from our scenario in which both teams adapt their strategies in the game.

When it comes to the multi-agent environments, multi-agent reinforcement learning (MARL) has been widely applied in various applications, such as transportation (Fernandez-Gauna et al., 2015), social sciences (Leibo et al., 2017), resource management (Hussin et al., 2015), and controlling a group of autonomous vehicles (Hung & Givigi, 2017). MARL has an issue of instability of the training process. To deal with the instability issue, MADDPG (Lowe et al., 2017) and M3DDPG (Li et al., 2019) have been proposed which adopt a centralized critic within the actor-critic learning framework to reduce the variance of policy gradient results. However, these methods have been designed for deterministic policies only. These methods do not perform as well in our baseball game scenario. Opponent modeling (Zhang & Lesser, 2010; Foerster et al., 2017b) is another method in which each agent can explore the opponent's strategy. Foerster et al. (Foerster et al., 2017b) propose a learning method, named Learning with Opponent-Learning Awareness (LOLA), to consider the learning processes of other agents. Their method has successfully enabled the cooperation of two players in repeated prisoner's dilemma games. In this paper, we further extend the discussion to the learning issue of multiple agents in a competitive game scenario.

## 3 MODIFIED DEEP CFR ALGORITHM & STRATEGY ADAPTATION

### 3.1 MODIFIED DEEP CFR ALGORITHM

To study the impact of initial strategy over the strategy adaptation mechanism, we propose an algorithm based on the modification of the Deep CFR algorithm to learn the strategies of the batter and the pitcher. The Deep CFR algorithm is a state-of-the-art technique to solve the imperfect information game, especially the Poker game, by traversing the game tree and playing with the regret matching algorithm at each iteration to reach the Epsilon-equilibrium (Brown et al., 2019; Zinkevich et al., 2007). However, due to the stochastic state transitions in our baseball game scenario, it is more efficient to learn the state-value function via temporal-difference (TD) learning (Sutton, 1988) than the tree searching method originally used in Deep CFR. Besides, baseball games are inherently highly uncertain and has only the final reward (win or lose). To simplify the problem, we learn the strategies of the pitcher and the batter in a half-inning and then apply the learned strategies for the whole game.

In a half-inning game, a state $s$ is represented by a 5-tuple vector: (runs, outs, runners on, strike, ball). The batter's final reward $V_t$ is defined as expressed in Equation 1, while the pitcher's final reward is defined as $-V_t$. We regard the final reward as "the state-value of the terminal state $V_t(s)$" to avoid the possible confusion with the term "reward", which is to be discussed later.

$$V_t(s) = \begin{cases} -20 \cdot \delta[runs] + 30 \cdot \delta[runs - 1] + 90 \cdot \delta[runs - 2] \\ +180 \cdot \delta[runs - 3] + 300 \cdot \delta[runs - 4] + 400 \cdot \delta[runs - 5] \\ \qquad\qquad\qquad\qquad\qquad\qquad if\ runs \leq 5 \\ 400 + (runs - 5) \cdot 30 \qquad\qquad if\ runs > 5 \end{cases} \tag{1}$$

The above definition is based on the observation that the average number of runs per game in MLB is about 5 and the intuition that different number of runs in a half-inning would have different impacts on the final winning percentage.

Our algorithm alternatively estimates the payoff matrices and updates the two agents' strategies based on the modified Deep CFR algorithm. In two-player zero-sum games, it has been proven that if both agents play according to CFR at every iteration, then these two agents' average

strategies will gradually converge to the Nash equilibrium as the number of iterations approaches infinity(Cesa-Bianchi & Lugosi, 2006; Zinkevich et al., 2007; Waugh, 2009).

In more details, at the Iteration t, given the batter's strategy $\sigma_b'^{t-1}$ , the pitcher's strategy $\sigma_p'^{t-1}$, and $V_t(s)$, a state-value network $V(s|\theta_v)$ is trained from scratch by using TD learning. Based on $V(s|\theta_v)$, we define the trained reward as

$$reward_{trained} = V' - V(s|\theta_v) \quad where \quad V' = \begin{cases} V(s'|\theta_v) & s' \neq s_{terminal} \\ V_t(s') & s' = s_{terminal} \end{cases} \tag{2}$$

where $s'$ is the next state, depending on the current state and the two agents' joint actions $(a_b, a_p)$.

Since this game only has one final reward, we manually choose a default reward $reward_{default}$ in order to improve the training efficiency. In our approach, the reward is defined as

$$reward = [V' - V(s|\theta_v)] \cdot \frac{t}{K_1} + reward_{default} \cdot (1 - \frac{t}{K_1}) \tag{3}$$

where $K_1$ is a manually selected constant. Based on the above definitions, an action-value network $Q(s, a_b, a_p|\theta_Q)$ is trained from scratch to estimate the expected reward of the two agents' joint actions $(a_b, a_p)$. We can express $Q(s, a_b, a_p|\theta_Q))$ at the state $s$ as a $5 \times 5$ payoff matrix with respect to pitcher's and batter's actions. With $Q(s, a_b, a_p|\theta_Q)$, the batter's instantaneous regrets $r_b$ can be estimated by the following equation:

$$r_b(s, a_b) = \mathbb{E}_{a_p \sim \sigma_p'^{t-1}(s)}[Q(s, a_b, a_p|\theta_Q)] - \mathbb{E}_{a_p \sim \sigma_p'^{t-1}(s), a_b \sim \sigma_b'^{t-1}(s)}[Q(s, a_b, a_p|\theta_Q)] \tag{4}$$

The pitcher's instantaneous regrets $r_p$ is calculated in a similar way. We assume both agents know each other's strategies in order to improve the learning efficiency. The sharing of strategy information is feasible in real life since we can ask the pitcher and the batter in the same team to compete with each other to learn their individual strategies.

These instantaneous regrets $r_b(s, a_b)$ and $r_p(s, a_p)$ are converted to the new strategies $\sigma_b^t, \sigma_p^t$ based on the following equation:

$$\sigma^t(s, a) = \frac{r_+(s, a)}{\Sigma_{a' \in A(s)} r_+(s, a')} \tag{5}$$

where $A(s)$ denotes the actions available at the state $s$ and $r_+(s, a) = \max(r(s, a), 0)$. If $\Sigma_{a' \in A(s)} r_+(s, a') = 0$, each action is assigned an equal probability.

Equation 5 comes from the modification of the Deep CFR algorithm. The original Deep CFR algorithm converts the agent's accumulated regret $R$ into the strategy by using the regret matching algorithm at each iteration (Hart & Mas-Colell, 2000; Zinkevich et al., 2007; Brown et al., 2019). Due to the heavy computational load of the learning process, we cannot afford too many iterations. Hence, we replace the accumulated regret $R$ in the Deep CFR algorithm by the instantaneous regrets $r$. However, the replacement of $R$ by $r$ results in larger strategy variations at different iterations. To mitigate this problem, we define the new strategy based on the following equation

$$\sigma'^t = \bar{\sigma}^{t-1} \cdot \frac{t}{K_2} + \sigma^t \cdot (1 - \frac{t}{K_2}) \tag{6}$$

where $\bar{\sigma}^{t-1}$ denotes the average strategy and $K_2$ is a manually selected constant to balance between $\bar{\sigma}^{t-1}$ and $\sigma^t$. At the next Iteration $(t + 1)$, $\sigma_b'^t, \sigma_p'^t$ are used to train a new value network $V(s|\theta_v)$. Meanwhile, $\sigma_b^t, \sigma_p^t$ are accumulated respectively in their strategy memories $M_\sigma$ as encountering the state $s$, weighted by $t$ as expressed below:

$$M_\sigma(s, a) \leftarrow M_\sigma(s, a) + \sigma^t(s, a) \cdot t \tag{7}$$

The average strategy at Iteration t is then computed from the strategy memories, expressed as

$$\bar{\sigma}^{t-1}(s, a) = \frac{M_\sigma(s, a)}{\Sigma_{a \in A(s)} M_\sigma(s, a)} \tag{8}$$

where $A(s)$ denotes the actions available at the state $s$. Note that the state-value function $V(s|\theta_v)$ highly depends on the agents' strategies, which have large variations at different iterations. Hence, $V(s|\theta_v)$ has to be retrained from scratch at every iteration.

### 3.2 STRATEGY ADAPTATION

At the beginning of the baseball game, each team plays against each other with its initial strategies $\sigma_p$ and $\sigma_b$ for the pitcher and the batter, respectively. The payoff matrix $Q(s, a_b, a_p | \theta_Q)$ at each state is then learned based on $\sigma_p$ and $\sigma_b$. The payoff matrix is used for a team to estimate the expected reward of the two agents' joint actions at any state. As the game proceeds, we propose a strategy adaptation mechanism to gradually modify the strategies. In the following paragraphs, we present the adaptation mechanism for the batter only. The adaptation mechanism for the pitcher can be deduced in an analogous way.

During the team's batting time, $\sigma_b$ is the batter's initial strategy and $\sigma_p$ is the prior assumption about the opponent pitcher's strategy. At each pitch, the batter anticipates what the pitcher will pitch based on the past observations over the pitcher's actions. In Equation 9, $O(s)$ denotes all the past observations at the state $s$ and $\tilde{\sigma}_p(s, a_p)$ denotes the predicted probability of the pitcher's action $a_p$ at the state $s$. In our mechanism, we treat $\tilde{\sigma}_p(s, a_p)$ as the posterior probability, conditioned on the past observations and the prior belief. That is,

$$\tilde{\sigma}_p(s, a_p) = \pi(a_p | s, O(s)) = \frac{\Pi_{i=1}^{N(O(s))} \pi(o_i(s) | s, a_p)}{\pi(O(s))} \sigma_p(s, a_p) \tag{9}$$

In Equation 9, $\sigma_p(s, a_p)$ denotes the prior knowledge about the pitcher's action at the state $s$. $\Pi_{i=1}^{N(O(s))} \pi(o_i(s) | s, a_p)$ denotes the likelihood function based on all the past observations at this state $s$ for the pitcher's action $a_p$. Here, $N(O(s))$ denotes the number of observations at the state $s$.

In addition to properly anticipate the pitcher's action, the batter should also know the expected reward of each action under different situations, which are expressed by $Q(s, a_b, a_p | \theta_Q)$. The batter can then obtain the advantageous strategy $\tilde{\sigma}_b$ by calculating the instantaneous regret in 4 based on $Q(s, a_b, a_p | \theta_Q), \tilde{\sigma}_p(s, a)$, and $\sigma_b(s, a)$. After that, he can update his strategy $\tilde{\sigma}_b$ based on 5. To gradually modify the strategy, the batter also takes into account the initial strategy $\sigma_b(s, a)$. The adapted strategy at the state $s$ is then defined as

$$\sigma_b^*(s, a_b) = \sigma_b(s, a_b) \cdot (1 - \frac{N(O(s))}{K_3}) + \tilde{\sigma}_b(s, a_b) \cdot \frac{N(O(s))}{K_3} \tag{10}$$

where $K_3$ is a manually selected constant. Equation 10 indicates that $\sigma_b^*(s, a_b)$ depends more on $\tilde{\sigma}_b$ as the number of observations $N(O(s))$ at this state $s$ increases. This is similar to the batter's behavior in real-life baseball games.

As the batter's strategy is adaptively modified as described above, the strategy adaptation of the opponent pitcher is performed based on the batter's reaction in an analogous way. As the number of observations $N(O(s))$ reaches a pre-selected threshold $N_{pitch}$, both $\sigma_b(s, a_b)$ and $\sigma_p(s, a_p)$ are updated based on the following equations:

$$\sigma_p(s, a_p) \leftarrow \sigma_p(s, a_p) \cdot (1 - \eta) + \tilde{\sigma}_p(s, a_p) \cdot \eta \tag{11}$$

$$\sigma_b(s, a_b) \leftarrow \sigma_b(s, a_b) \cdot (1 - \eta) + \tilde{\sigma}_b(s, a_b) \cdot \eta \tag{12}$$

where $\eta$ is a manually determined learning rate. After the update, all the past observations are reset and the same strategy adaptation process starts again.

In our simulation, each team has its own strategy pair $(\sigma_p, \sigma_b)$ for its pitcher and its batter. For each team, either the pitcher's strategy or the batter's strategy is adaptively modified depending on whether the team is under pitching or batting. In our baseball game scenario, each team plays best-of-three games with every other team. That is, each team has at most three games to observe the opponent team.

# 4 Experimental results

## 4.1 The Strategy learned from the modified Deep CFR algorithm

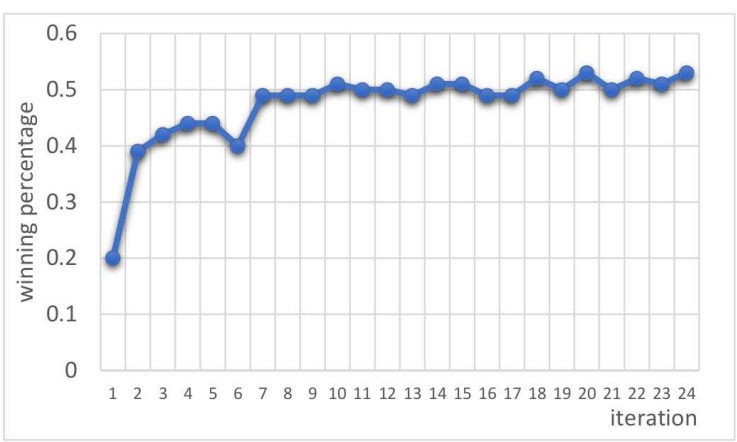

Figure 3: The winning percentage of the learned strategies in the training process.

We train the pitcher and batter in a half-inning game by the modified Deep CFR algorithm, then we apply the learned strategy to the whole(nine-inning) game. At each iteration of the training process, the trained team plays the best-of-three game against every team of the Team-0 to Team-12 (listed on Table 3) for 200 series. The averaged winning percentage continuously increases as the iteration proceeds, as shown in Figure 3. This implies that the learned strategy is getting closer to the Nash-equilibrium strategy if compared with the strategies of the other teams.

Table 2 lists the learned strategies for the pitcher and the batter at 24 different states based on the modified Deep CFR algorithm. Here, we attempt to observe the strategy differences with respect to different states. In Table 2, the "count" denotes the number of balls and the number of strikes. For example, (2-1) means 2 balls and 1 strike. Each row in Table 2 represents the probability of each action with respect to a specific count for the pitcher and the batter, respectively. It is very interesting to observe that the learned strategy at some state is actually quite similar to real-life baseball strategies. For example,

Case 1: Count of two strikes
When there are already two strikes, the batter has less freedom to sit on any preferred zone or specific pitch type. As shown in Table 2, for the batter, the probability of "any" is indeed quite high for these 2-strike cases, except the very special case of 3-2 (3 balls, 2 strikes). On the other hand, for the pitcher, the probability of curveC becomes higher because the pitcher has more freedom to pitch out of the strike zone to fool the batter.

Case 2: Count of 3-1
For this case (3 balls, 1 strike), we may consider two different situations. When the number of runners is smaller than or equal to 1 (Situation 1), the probability of fastM is quite high for the pitcher (0.98) because the pitcher wants to avoid a possible walk in this situation. On the other hand, the batter would have the freedom to reduce the attack zone to make a solid contact (the probability of (fastH+any) in Situation 1 is only 0.27 for the batter). However, when the number of runners are larger than 1 (Situation 2), a batter will be more aggressive since it is a good opportunity to score (the probability of (fastH+any) in Situation 2 for the batter is 0.44). On the other hand, the pitcher needs to pitch more carefully to prevent solid contact. (the probability of fastM (0.76) is lower than the probability (0.98) in Situation 1).

Table 2: Strategies of the batter and the pitcher.

Situation1: number of runners$\leq$ 1. Situation2: number of runners $>$ 1.

| | Count | \multicolumn{5}{c}{Batter's strategy} | Count | \multicolumn{5}{c}{Pitcher's strategy} |

| | Count | wait | fastM | fastH | curveH | any | Count | fastM | fastS | curveM | curveS | curveC |
|---|---|---|---|---|---|---|---|---|---|---|---|---|
| **Situation1** | 0-0 | 0.25 | 0.28 | 0.47 | 0.00 | 0.00 | 0-0 | 0.27 | 0.25 | 0.33 | 0.12 | 0.04 |
| | 1-0 | 0.18 | 0.21 | 0.52 | 0.09 | 0.00 | 1-0 | 0.23 | 0.27 | 0.32 | 0.16 | 0.02 |
| | 0-1 | 0.20 | 0.27 | 0.49 | 0.04 | 0.00 | 0-1 | 0.31 | 0.27 | 0.32 | 0.08 | 0.03 |
| | 1-1 | 0.21 | 0.28 | 0.47 | 0.04 | 0.00 | 1-1 | 0.28 | 0.33 | 0.28 | 0.09 | 0.02 |
| | 2-1 | 0.29 | 0.24 | 0.43 | 0.04 | 0.00 | 2-1 | 0.19 | 0.27 | 0.34 | 0.18 | 0.02 |
| | 3-1 | 0.28 | 0.19 | 0.20 | 0.26 | 0.07 | 3-1 | 0.98 | 0.00 | 0.02 | 0.00 | 0.00 |
| | 2-0 | 0.19 | 0.16 | 0.49 | 0.17 | 0.00 | 2-0 | 0.28 | 0.27 | 0.24 | 0.18 | 0.03 |
| | 3-0 | 0.30 | 0.24 | 0.18 | 0.23 | 0.05 | 3-0 | 0.98 | 0.00 | 0.02 | 0.00 | 0.00 |
| | 0-2 | 0.00 | 0.00 | 0.08 | 0.03 | 0.89 | 0-2 | 0.30 | 0.06 | 0.34 | 0.19 | 0.11 |
| | 1-2 | 0.00 | 0.02 | 0.18 | 0.02 | 0.78 | 1-2 | 0.36 | 0.06 | 0.33 | 0.18 | 0.07 |
| | 2-2 | 0.00 | 0.00 | 0.15 | 0.03 | 0.82 | 2-2 | 0.33 | 0.05 | 0.30 | 0.22 | 0.10 |
| | 3-2 | 0.04 | 0.04 | 0.54 | 0.06 | 0.32 | 3-2 | 0.68 | 0.12 | 0.16 | 0.05 | 0.00 |
| | Count | wait | fastM | fastH | curveH | any | Count | fastM | fastS | curveM | curveS | curveC |
| **Situation2** | 0-0 | 0.15 | 0.16 | 0.49 | 0.10 | 0.11 | 0-0 | 0.17 | 0.25 | 0.32 | 0.19 | 0.07 |
| | 1-0 | 0.14 | 0.13 | 0.43 | 0.17 | 0.13 | 1-0 | 0.17 | 0.24 | 0.30 | 0.22 | 0.07 |
| | 0-1 | 0.10 | 0.11 | 0.60 | 0.10 | 0.10 | 0-1 | 0.13 | 0.22 | 0.40 | 0.22 | 0.03 |
| | 1-1 | 0.14 | 0.14 | 0.48 | 0.16 | 0.08 | 1-1 | 0.13 | 0.30 | 0.29 | 0.23 | 0.05 |
| | 2-1 | 0.13 | 0.12 | 0.46 | 0.18 | 0.12 | 2-1 | 0.17 | 0.19 | 0.30 | 0.28 | 0.06 |
| | 3-1 | 0.14 | 0.29 | 0.34 | 0.14 | 0.10 | 3-1 | 0.76 | 0.14 | 0.06 | 0.04 | 0.01 |
| | 2-0 | 0.12 | 0.10 | 0.46 | 0.18 | 0.13 | 2-0 | 0.13 | 0.21 | 0.31 | 0.26 | 0.08 |
| | 3-0 | 0.17 | 0.41 | 0.24 | 0.11 | 0.07 | 3-0 | 0.71 | 0.16 | 0.08 | 0.04 | 0.01 |
| | 0-2 | 0.00 | 0.00 | 0.04 | 0.02 | 0.95 | 0-2 | 0.24 | 0.05 | 0.31 | 0.22 | 0.19 |
| | 1-2 | 0.00 | 0.01 | 0.06 | 0.07 | 0.87 | 1-2 | 0.21 | 0.04 | 0.30 | 0.23 | 0.23 |
| | 2-2 | 0.00 | 0.01 | 0.11 | 0.10 | 0.78 | 2-2 | 0.29 | 0.09 | 0.29 | 0.15 | 0.18 |
| | 3-2 | 0.01 | 0.04 | 0.55 | 0.11 | 0.28 | 3-2 | 0.40 | 0.20 | 0.19 | 0.17 | 0.05 |

## 4.2 STRATEGY ADAPTATION

In our simulation of competitive games, we form 14 teams, including 13 manually determined teams and 1 team trained by the modified Deep CFR algorithm. Each team has its own initial strategy and the payoff matrix. Each team plays the best-of-three games against every other team for 200 series. In each series, each team has its own initial strategy at the beginning, followed by the adaptation mechanism mentioned in Section 3.2 to update its strategy based on the observations of its opponent.

In the left three columns of Table 3, we list the 14 teams, together with the characteristics of the pitcher and the batter in each team. For Team-0 to Team-3, an active batter is more aggressive to swing, while a passive batter tends to wait or to choose the action fastM more often. On the other hand, an active pitcher tends to pitch to the middle, while a passive pitcher tends to pitch to the corner. For Team-4 to Team-7, both the pitcher and the batter keep choosing the same action. For example, the batter of Team-4 always chooses the action fastH and the pitcher always chooses the action fastS. Team-8 to Team-11 are intentionally formed to defeat Team-4 to Team-8. For example, the strategy of Team-8 is especially designed to defeat Team-6, while the strategy of Team-9 is designed to defeat Team-4. The strategies of Team 0 to Team 3 & Team 8 to Team 11 are listed in B.

In the two columns on the right of Table 3, we list the averaged winning percentage (WP) of each team with respect to the other teams. For the teams with "specific tendency" (Team-4 to Team-11), some of them have higher WP, such as Team 5 and Team 9 in the without-adaptation domain. This indicates some strategies, like always pitch to the corner, can be very effective in winning the games. However, as the strategy adaptation mechanism is employed, the WP of most teams with "specific tendency" actually decreases. This is quite reasonable since the team with "specific tendency" strategy will restrict themselves to properly modify their strategies against various kinds of opponents and can be easily exploited by their opponents. On the other hand, it seems the the strategy learned from the modified Deep CFR algorithm for Team-13 can serve as a good initial strategy if we want to adopt the strategy adaptation mechanism.

In Table 4, we compare the winning percentage among the first four teams (Team-0 to Team-3) in the without-adaptation domain. The simulation results are very similar to real-life baseball games: (1)

Table 3: Average Winning Percentage (WP) for each team in the without-adaptation domain (non) and the with-adaptation domain (adap.).

| | The batter | The pitcher | WP(non) | WP(adap.) |
|---|---|---|---|---|
| Team 0 | Active | Active | 0.55 | 0.61 |
| Team 1 | Active | Passive | 0.45 | 0.49 |
| Team 2 | Passive | Active | 0.52 | 0.50 |
| Team 3 | Passive | Passive | 0.41 | 0.55 |
| Team 4 | fastH | fastS | 0.57 | 0.52 |
| Team 5 | fastH | curveM | 0.60 | 0.41 |
| Team 6 | any | fastS | 0.43 | 0.53 |
| Team 7 | any | curveM | 0.49 | 0.30 |
| Team 8 | Exploit fastS P | Exploit any B | 0.59 | 0.46 |
| Team 9 | Exploit fastS P | Exploit fastH B | 0.64 | 0.41 |
| Team 10 | Exploit curveM P | Exploit any B | 0.48 | 0.52 |
| Team 11 | Exploit curveM P | Exploit fastH B | 0.53 | 0.45 |
| Team 12 | random | random | 0.20 | 0.48 |
| Team 13 | Trained with modified Deep CFR | | 0.54 | 0.71 |

Table 4: Winning Percentage (WP) among Team-0 to Team-3. The number stands for the WP of the corresponding team listed in the left column.

| | Team 0 | Team 1 | Team 2 | Team 3 | Team12 |
|---|---|---|---|---|---|
| Team 0 | --- | 0.47 | 0.67 | 0.61 | 0.93 |
| Team 1 | 0.53 | --- | 0.50 | 0.48 | 0.73 |
| Team 2 | 0.33 | 0.50 | --- | 0.66 | 0.90 |
| Team 3 | 0.39 | 0.52 | 0.34 | --- | 0.66 |

when the batter is active, a passive pitcher has a better chance to win than an active pitcher (Team-1 vs Team-0); (2) when the batter is passive, an active pitcher has a better chance to win than a passive pitcher (Team-2 vs Team-3); (3) when the pitcher is active, an active batter has a better chance to win than a passive batter (Team-0 vs Team-2); and (4) when the pitcher is passive, a passive batter has a better chance to win than an active batter (Team-3 vs Team-1).

Table 4 shows that none of these four teams can dominate the games. Team-1 beats Team-0, Team-0 beats Team-2, Team-2 beats Team-3, while Team-3 beats Team-1. This implies the importance of strategy adaptation. Besides, in Table 4, we also show that all these four teams have better chances to beat Team-12, which adopts a random initial strategy. This implies that a team with a strategy is better than a team without any strategy.

When two teams, if named as home team and guest team, compete with each other, there are four possible combinations in strategy adaptation: (1) only the guest team adapts, (2) both teams do not adapt, (3) both teams adapt, and (4) only the home team adapts. In Table 5, we list the competition results of these four different combinations when we choose Team-5 or Team-13 as the home team and choose each of the remaining teams as the guest team. Here, Team-5 represents those teams with a "specific tendency" strategy, while Team-13 represents the team with an initial strategy learned from the modified Deep CFR algorithm. Besides, in Table 5, we classify Team-4 to Team-11 as Category 1, whose initial strategy has "specific tendency", while classify the other teams as Category 2. On the right three columns of Table 5, "avg" represents the averaged WP value with respect to all the guest teams, "avg-1" represents the averaged WP with respect to Category-1 teams, and "avg-2" represents the averaged WP value with respect to Category-2 teams.

In Table 5, we observe some interesting phenomena:

1. In terms of "avg", Team-5 has the best performance when both teams do not adapt, while has the worst performance when both teams adapt.
2. In terms of "avg1", the WP of Team-5 is roughly 50%, basically independent of the adaptation

mechanism.

3. In terms of "avg2", the WP of Team-5 drops drastically from "No Adapt" to "Both Adapt".

4. For Team-13, the WP increases if the strategy adaptation mechanism is adopted (see "Both Adapt" and "Home Adapt").

5. With strategy adaptation, the WP of Team-13 against any other team is higher than 50 %. This implies, in average, Team-13 always wins when strategy adaptation is used.

Based on the above observations, we have the following conclusions:

**1. Those teams with a "specific tendency" strategy might have high winning percentage against some of the other teams when there is no strategy adaptation. However, as the adaptation mechanism is adopted, those teams' advantage tends to drop.**

**2. The team with an initial strategy learned from the modified Deep CFR algorithm benefits from the strategy adaptation mechanism.**

Table 5: WP of Team-5 and Team-13 against the other teams for 2000 series.

WP of Team 5 against the other teams.

| domain | 0 | 1 | 2 | 3 | 4 | 5 | 6 | 7 | 8 | 9 | 10 | 11 | 12 | 13 | avg | avg1 | avg2 |
|---|---|---|---|---|---|---|---|---|---|---|---|---|---|---|---|---|---|
| Guest Adapt | 0.69 | 0.56 | 0.76 | 0.57 | 0.78 | 0.39 | 0.93 | 0.27 | 0.70 | 0.08 | 0.77 | 0.30 | 0.84 | 0.35 | 0.57 | 0.53 | 0.63 |
| No Adapt | 0.75 | 0.77 | 0.66 | 0.69 | 0.98 | 0.49 | 0.80 | 0.10 | 0.83 | 0.38 | 0.34 | 0.05 | 0.93 | 0.70 | 0.61 | 0.50 | 0.75 |
| Both Adapt | 0.27 | 0.65 | 0.24 | 0.37 | 0.72 | 0.50 | 0.43 | 0.41 | 0.43 | 0.78 | 0.13 | 0.27 | 0.28 | 0.29 | 0.41 | 0.46 | 0.35 |
| Home Adapt | 0.17 | 0.54 | 0.24 | 0.62 | 0.39 | 0.59 | 0.42 | 0.61 | 0.21 | 0.36 | 0.51 | 0.67 | 0.87 | 0.38 | 0.47 | 0.47 | 0.47 |

WP of Team 13 against the other teams.

| domain | 0 | 1 | 2 | 3 | 4 | 5 | 6 | 7 | 8 | 9 | 10 | 11 | 12 | 13 | avg | avg1 | avg2 |
|---|---|---|---|---|---|---|---|---|---|---|---|---|---|---|---|---|---|
| Guest Adapt | 0.37 | 0.43 | 0.63 | 0.53 | 0.43 | 0.63 | 0.61 | 0.76 | 0.62 | 0.50 | 0.61 | 0.69 | 0.77 | 0.31 | 0.56 | 0.61 | 0.51 |
| No Adapt | 0.46 | 0.57 | 0.57 | 0.67 | 0.47 | 0.31 | 0.65 | 0.46 | 0.44 | 0.32 | 0.67 | 0.57 | 0.90 | 0.50 | 0.54 | 0.49 | 0.61 |
| Both Adapt | 0.62 | 0.58 | 0.77 | 0.66 | 0.71 | 0.72 | 0.80 | 0.84 | 0.75 | 0.58 | 0.82 | 0.76 | 0.88 | 0.49 | 0.71 | 0.75 | 0.67 |
| Home Adapt | 0.71 | 0.74 | 0.71 | 0.74 | 0.77 | 0.67 | 0.72 | 0.61 | 0.75 | 0.62 | 0.72 | 0.56 | 0.94 | 0.71 | 0.71 | 0.68 | 0.76 |

## 5 CONCLUSION

In this paper, we construct a simplified baseball game scenario to develop and evaluate the adaptation capability of learning agents. We are especially interested in what kinds of teams have a better chance to survive when there is strategy adaptation. We propose a modified Deep CFR algorithm to learn an initial strategies of the batter and pitcher. The experimental results indeed show that the team with an initial strategy learned from the modified Deep CFR algorithm is more favorable than a team with a deterministic initial strategy in our baseball game scenario.

In this work, since we only focus on the impact of strategies on Winning Percentage, the capabilities of the pitcher and the batter are fixed across different teams. In the future work, we would relax this constraint and both teams are required to anticipate the " capabilities and strategies " of the opponents during the game process. This setup will make the game scenario more realistic, and the behavior of the agents would be more similar to the behavior of human players in real life.

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

# A PSEUDO CODE

## A.1 MODIFIED DEEP CFR

---
**Algorithm 1** Modified Deep CFR
---
1: **function** LEARNSTRATEGY
2:     Initialize strategies $\sigma_b^0, \sigma_p^0, \sigma_b'^0, \sigma_p'^0$ with random strategies.
3:     Initialize Strategy memories $M_{\sigma_b}, M_{\sigma_p}$.
4:     Choose final reward $V_t(s)$ and default reward $reward_{default}$. Select constant $K_1$ and $K_2$.
5:     **for** Iteration t=1 to T **do**
6:         $V(s|\theta_V), M_{\sigma_b}, M_{\sigma_p} = \text{LEARNVALUE}(\sigma_b^{t-1}, \sigma_p^{t-1}, \sigma_b'^{t-1}, \sigma_p'^{t-1}, M_{\sigma_b}, M_{\sigma_p})$
7:         $Q(s, a_b, a_p|\theta_Q) = \text{LEARNQ}(V(s|\theta_V)$
8:         Calculate both players' instantaneous regrets $r_b, r_p$ respectively with equation 4.
9:         Convert the both players' instantaneous regrets $r_b, r_p$ to the new strategies $\sigma_b^t, \sigma_p^t$ respectively with equation 5.
10:        Calculate the average strategy $\bar{\sigma}_b^{t-1}, \bar{\sigma}_p^{t-1}$ from Strategy memories $M_{\sigma_b}, M_{\sigma_p}$ respectively with equation 8.
11:        Calculate the new strategies $\sigma_b'^t, \sigma_p'^t$ with equation 6.
12:     **end for**
13:     **return** $\bar{\sigma}_b^T, \bar{\sigma}_p^T$
14: **end function**

---
**Algorithm 2** Learning Value function
---
1: **function** LEARNVALUE($\sigma_b, \sigma_p, \sigma_b', \sigma_p', M_{\sigma_b}, M_{\sigma_p}$)
2:     Initialize $V(s|\theta_V)$, Replay buffer $D$.
3:     **for** episode t=1 to K **do**
4:         Initialize $s$.
5:         **repeat** (for each state of episode)
6:             Accumulate $\sigma_b(s), \sigma_p(s)$ respectively in their strategy memories, weighted by $t$ with equation 7.
7:             Select actions according to $\sigma_b', \sigma_p'$ from $s$ and observe the new state $s'$.
8:             Store transition $(s, s')$ Into $D$.
9:             Sample a random minibatch of $N$ transitions $(s_i, s_i')$ from $D$. Update the value function $V(s|\theta_V)$ by minimizing the loss:
$$L = \frac{1}{N}\Sigma_i[(V(s_i|\theta_V) - V')^2] \quad \text{where} \quad V' = \begin{cases} V(s_i'|\theta_v) & s_i' \neq s_{terminal} \\ V_t(s_i') & s_i' = s_{terminal} \end{cases}$$
10:             $s \leftarrow s'$
11:         **until** $s$ is terminal
12:     **end for**
13:     **return** $V(s|\theta_V), M_{\sigma_b}, M_{\sigma_p}$
14: **end function**

---

In our experiment, The hyperparameters are chosen in the following: $K_1 = 30, K_2 = 30, K_3 = 20, N_{pitch} = 4, \eta = 0.1$.

---

**Algorithm 3** Learning Q function

---
1: **function** LEARNQ($V(s|\theta_V)$)
2:     Initialize $Q(s, a_b, a_p|\theta_Q)$, Replay buffer $D$.
3:     **for** episode t=1 to K **do**
4:         Randomly initialize $s$.
5:         **repeat** (for each state of episode)
6:             Randomly select actions $(a_b, a_p)$ from $s$ and observe the new state $s'$ and $reward$
            according to equation 3.
7:             Store transition $(s, a_b, a_p, reward, s')$ into $D$.
8:             Sample a random minibatch of $N$ transitions $(s_i, a_{b,i}, a_{p,i}, reward_i, s'_i)$ from $D$.
            Update the $Q$ function $Q(s, a_b, a_p|\theta_Q)$ by minimizing the loss:
            $L = \frac{1}{N}\Sigma_i[(Q(s_i, a_{b,i}, a_{p,i}|\theta_Q) - reward_i)^2]$
9:             $s \leftarrow s'$
10:         **until** $s$ is terminal
11:     **end for**
12:     **return** $Q(s, a_b, a_p|\theta_Q)$
13: **end function**

---

## A.2 ADAPTATION MECHANISM

---

**Algorithm 4** Adaptation mechanism for the batter during a game

---
1: **procedure** A GAME
2:     Set $N_{pitch}, \eta, K_3$.
3:     Initialize the observation memory $O$.
4:     Prepare $\sigma_b$(The batter's initial strategy), $\sigma_p$(the prior assumption about the opponent pitcher's strategy), and payoff matrix $Q(s, a_b, a_p|\theta_Q)$.
5:     **repeat** (for each state of episode)
6:         **if** $N(O(s)) \geq 1$ **then**       // $N(O(s))$ denotes the number of observations at the state $s$.
7:             Anticipate the pitcher's strategy $\tilde{\sigma}_p(s, a)$ based on the past observations in $O(s)$ according to equation 9.
8:             Calculate the regret $r_b(s, a_b)$ by 4 with $Q(s, a_b, a_p|\theta_Q), \tilde{\sigma}_p(s, a), \sigma_b(s, a)$.
9:             Convert $r_b(s, a_b)$ to the strategy $\tilde{\sigma}_b(s, a)$ with equation 5.
10:         **end if**
11:         Calculate $\sigma_b^*(s, a_b)$ according to equation 10.
12:         We select action according to $\sigma_b^*(s, a_b)$, and the opponent pitcher select his action.
13:         The game proceeds to the new state $s'$.
14:         Restore the observed pitch into $O(s)$.
15:         $N(O(s)) += 1$.
16:         **if** $N(O(s)) == N_{pitch}$ **then**
17:             update $\sigma_p(s, a_p)$ by equation 11
18:             update $\sigma_b(s, a_b)$ by equation 12
19:             $N(O(s)) = 0$
20:             Reset $O(s)$.
21:         **end if**
22:         $s \leftarrow s'$
23:     **until** $s$ is terminal
24: **end procedure**

---

# B THE STRATEGIES OF TEAM 0 TO TEAM 3, AND TEAM 8 TO TEAM 11

During a game, these team's strategies only depend on "count".

### Table 6: The strategy of Team 0

| | Batter's strategy | | | | | | Pitcher's strategy | | | | |
|---|---|---|---|---|---|---|---|---|---|---|---|
| Count | wait | fastM | fastH | curveH | any | Count | fastM | fastS | curveM | curveS | curveC |
| 0-0 | 0.00 | 0.60 | 0.30 | 0.10 | 0.00 | 0-0 | 0.60 | 0.30 | 0.10 | 0.00 | 0.00 |
| 1-0 | 0.00 | 0.60 | 0.30 | 0.10 | 0.00 | 1-0 | 0.60 | 0.30 | 0.10 | 0.00 | 0.00 |
| 0-1 | 0.00 | 0.00 | 0.70 | 0.00 | 0.30 | 0-1 | 0.30 | 0.60 | 0.00 | 0.10 | 0.00 |
| 1-1 | 0.00 | 0.00 | 0.70 | 0.00 | 0.30 | 1-1 | 0.30 | 0.60 | 0.00 | 0.10 | 0.00 |
| 2-1 | 0.00 | 0.00 | 0.70 | 0.00 | 0.30 | 2-1 | 0.30 | 0.60 | 0.00 | 0.10 | 0.00 |
| 3-1 | 0.00 | 0.60 | 0.40 | 0.00 | 0.00 | 3-1 | 0.90 | 0.10 | 0.00 | 0.00 | 0.00 |
| 2-0 | 0.00 | 0.60 | 0.40 | 0.00 | 0.00 | 2-0 | 0.90 | 0.10 | 0.00 | 0.00 | 0.00 |
| 3-0 | 0.60 | 0.30 | 0.10 | 0.00 | 0.00 | 3-0 | 1.00 | 0.00 | 0.00 | 0.00 | 0.00 |
| 0-2 | 0.00 | 0.00 | 0.00 | 0.00 | 1.00 | 0-2 | 0.00 | 0.20 | 0.10 | 0.50 | 0.20 |
| 1-2 | 0.00 | 0.00 | 0.00 | 0.00 | 1.00 | 1-2 | 0.00 | 0.20 | 0.10 | 0.50 | 0.20 |
| 2-2 | 0.00 | 0.00 | 0.00 | 0.00 | 1.00 | 2-2 | 0.40 | 0.30 | 0.10 | 0.20 | 0.00 |
| 3-2 | 0.00 | 0.00 | 0.00 | 0.00 | 1.00 | 3-2 | 0.70 | 0.20 | 0.10 | 0.00 | 0.00 |

### Table 7: The strategy of Team 1

| | Batter's strategy | | | | | | Pitcher's strategy | | | | |
|---|---|---|---|---|---|---|---|---|---|---|---|
| Count | wait | fastM | fastH | curveH | any | Count | fastM | fastS | curveM | curveS | curveC |
| 0-0 | 0.00 | 0.60 | 0.30 | 0.10 | 0.00 | 0-0 | 0.00 | 0.50 | 0.40 | 0.10 | 0.00 |
| 1-0 | 0.00 | 0.60 | 0.30 | 0.10 | 0.00 | 1-0 | 0.00 | 0.50 | 0.40 | 0.10 | 0.00 |
| 0-1 | 0.00 | 0.00 | 0.70 | 0.00 | 0.30 | 0-1 | 0.00 | 0.20 | 0.10 | 0.50 | 0.20 |
| 1-1 | 0.00 | 0.00 | 0.70 | 0.00 | 0.30 | 1-1 | 0.00 | 0.20 | 0.10 | 0.50 | 0.20 |
| 2-1 | 0.00 | 0.00 | 0.70 | 0.00 | 0.30 | 2-1 | 0.00 | 0.20 | 0.10 | 0.50 | 0.20 |
| 3-1 | 0.00 | 0.60 | 0.40 | 0.00 | 0.00 | 3-1 | 0.10 | 0.40 | 0.30 | 0.20 | 0.00 |
| 2-0 | 0.00 | 0.60 | 0.40 | 0.00 | 0.00 | 2-0 | 0.10 | 0.40 | 0.30 | 0.20 | 0.00 |
| 3-0 | 0.60 | 0.30 | 0.10 | 0.00 | 0.00 | 3-0 | 0.50 | 0.30 | 0.20 | 0.00 | 0.00 |
| 0-2 | 0.00 | 0.00 | 0.00 | 0.00 | 1.00 | 0-2 | 0.00 | 0.10 | 0.00 | 0.20 | 0.70 |
| 1-2 | 0.00 | 0.00 | 0.00 | 0.00 | 1.00 | 1-2 | 0.00 | 0.10 | 0.00 | 0.20 | 0.70 |
| 2-2 | 0.00 | 0.00 | 0.00 | 0.00 | 1.00 | 2-2 | 0.00 | 0.20 | 0.10 | 0.40 | 0.30 |
| 3-2 | 0.00 | 0.00 | 0.00 | 0.00 | 1.00 | 3-2 | 0.20 | 0.30 | 0.20 | 0.30 | 0.00 |

### Table 8: The strategy of Team 2

| | Batter's strategy | | | | | | Pitcher's strategy | | | | |
|---|---|---|---|---|---|---|---|---|---|---|---|
| Count | wait | fastM | fastH | curveH | any | Count | fastM | fastS | curveM | curveS | curveC |
| 0-0 | 0.00 | 0.10 | 0.40 | 0.30 | 0.20 | 0-0 | 0.60 | 0.30 | 0.10 | 0.00 | 0.00 |
| 1-0 | 0.00 | 0.10 | 0.40 | 0.30 | 0.20 | 1-0 | 0.60 | 0.30 | 0.10 | 0.00 | 0.00 |
| 0-1 | 0.10 | 0.60 | 0.20 | 0.10 | 0.00 | 0-1 | 0.30 | 0.60 | 0.00 | 0.10 | 0.00 |
| 1-1 | 0.10 | 0.60 | 0.20 | 0.10 | 0.00 | 1-1 | 0.30 | 0.60 | 0.00 | 0.10 | 0.00 |
| 2-1 | 0.10 | 0.60 | 0.20 | 0.10 | 0.00 | 2-1 | 0.30 | 0.60 | 0.00 | 0.10 | 0.00 |
| 3-1 | 0.30 | 0.60 | 0.10 | 0.00 | 0.00 | 3-1 | 0.90 | 0.10 | 0.00 | 0.00 | 0.00 |
| 2-0 | 0.30 | 0.60 | 0.10 | 0.00 | 0.00 | 2-0 | 0.90 | 0.10 | 0.00 | 0.00 | 0.00 |
| 3-0 | 0.90 | 0.10 | 0.00 | 0.00 | 0.00 | 3-0 | 1.00 | 0.00 | 0.00 | 0.00 | 0.00 |
| 0-2 | 0.00 | 0.00 | 0.00 | 0.30 | 0.70 | 0-2 | 0.00 | 0.20 | 0.10 | 0.50 | 0.20 |
| 1-2 | 0.00 | 0.00 | 0.00 | 0.30 | 0.70 | 1-2 | 0.00 | 0.20 | 0.10 | 0.50 | 0.20 |
| 2-2 | 0.00 | 0.00 | 0.00 | 0.00 | 1.00 | 2-2 | 0.40 | 0.30 | 0.10 | 0.20 | 0.00 |
| 3-2 | 0.00 | 0.20 | 0.00 | 0.00 | 0.80 | 3-2 | 0.70 | 0.20 | 0.10 | 0.00 | 0.00 |

### Table 9: The strategy of Team 3

| | Batter's strategy | | | | | | Pitcher's strategy | | | | |
|---|---|---|---|---|---|---|---|---|---|---|---|
| Count | wait | fastM | fastH | curveH | any | Count | fastM | fastS | curveM | curveS | curveC |
| 0-0 | 0.00 | 0.10 | 0.40 | 0.30 | 0.20 | 0-0 | 0.00 | 0.50 | 0.40 | 0.10 | 0.00 |
| 1-0 | 0.00 | 0.10 | 0.40 | 0.30 | 0.20 | 1-0 | 0.00 | 0.50 | 0.40 | 0.10 | 0.00 |
| 0-1 | 0.10 | 0.60 | 0.20 | 0.10 | 0.00 | 0-1 | 0.00 | 0.20 | 0.10 | 0.50 | 0.20 |
| 1-1 | 0.10 | 0.60 | 0.20 | 0.10 | 0.00 | 1-1 | 0.00 | 0.20 | 0.10 | 0.50 | 0.20 |
| 2-1 | 0.10 | 0.60 | 0.20 | 0.10 | 0.00 | 2-1 | 0.00 | 0.20 | 0.10 | 0.50 | 0.20 |
| 3-1 | 0.30 | 0.60 | 0.10 | 0.00 | 0.00 | 3-1 | 0.10 | 0.40 | 0.30 | 0.20 | 0.00 |
| 2-0 | 0.30 | 0.60 | 0.10 | 0.00 | 0.00 | 2-0 | 0.10 | 0.40 | 0.30 | 0.20 | 0.00 |
| 3-0 | 0.90 | 0.10 | 0.00 | 0.00 | 0.00 | 3-0 | 0.50 | 0.30 | 0.20 | 0.00 | 0.00 |
| 0-2 | 0.00 | 0.00 | 0.00 | 0.30 | 0.70 | 0-2 | 0.00 | 0.10 | 0.00 | 0.20 | 0.70 |
| 1-2 | 0.00 | 0.00 | 0.00 | 0.30 | 0.70 | 1-2 | 0.00 | 0.10 | 0.00 | 0.20 | 0.70 |
| 2-2 | 0.00 | 0.00 | 0.00 | 0.00 | 1.00 | 2-2 | 0.00 | 0.20 | 0.10 | 0.40 | 0.30 |
| 3-2 | 0.00 | 0.20 | 0.00 | 0.00 | 0.80 | 3-2 | 0.20 | 0.30 | 0.20 | 0.30 | 0.00 |

### Table 10: The strategy of Team 8

| | Batter's strategy | | | | | | Pitcher's strategy | | | | |
|---|---|---|---|---|---|---|---|---|---|---|---|
| Count | wait | fastM | fastH | curveH | any | Count | fastM | fastS | curveM | curveS | curveC |
| 0-0 | 0.00 | 0.30 | 0.70 | 0.00 | 0.00 | 0-0 | 0.30 | 0.70 | 0.00 | 0.00 | 0.00 |
| 1-0 | 0.00 | 0.30 | 0.70 | 0.00 | 0.00 | 1-0 | 0.30 | 0.70 | 0.00 | 0.00 | 0.00 |
| 0-1 | 0.00 | 0.00 | 1.00 | 0.00 | 0.00 | 0-1 | 0.00 | 0.50 | 0.00 | 0.20 | 0.30 |
| 1-1 | 0.00 | 0.00 | 1.00 | 0.00 | 0.00 | 1-1 | 0.00 | 0.50 | 0.00 | 0.20 | 0.30 |
| 2-1 | 0.00 | 0.00 | 1.00 | 0.00 | 0.00 | 2-1 | 0.00 | 0.50 | 0.00 | 0.20 | 0.30 |
| 3-1 | 0.00 | 0.60 | 0.40 | 0.00 | 0.00 | 3-1 | 0.40 | 0.60 | 0.00 | 0.00 | 0.00 |
| 2-0 | 0.00 | 0.60 | 0.40 | 0.00 | 0.00 | 2-0 | 0.40 | 0.60 | 0.00 | 0.00 | 0.00 |
| 3-0 | 0.70 | 0.30 | 0.00 | 0.00 | 0.00 | 3-0 | 0.70 | 0.30 | 0.00 | 0.00 | 0.00 |
| 0-2 | 0.00 | 0.00 | 0.00 | 0.00 | 1.00 | 0-2 | 0.00 | 0.00 | 0.00 | 0.20 | 0.80 |
| 1-2 | 0.00 | 0.00 | 0.00 | 0.00 | 1.00 | 1-2 | 0.00 | 0.00 | 0.00 | 0.20 | 0.80 |
| 2-2 | 0.00 | 0.00 | 0.00 | 0.00 | 1.00 | 2-2 | 0.20 | 0.20 | 0.00 | 0.40 | 0.20 |
| 3-2 | 0.00 | 0.00 | 0.00 | 0.00 | 1.00 | 3-2 | 0.40 | 0.00 | 0.60 | 0.00 | 0.00 |

### Table 11: The strategy of Team 9

| | Batter's strategy | | | | | | Pitcher's strategy | | | | |
|---|---|---|---|---|---|---|---|---|---|---|---|
| Count | wait | fastM | fastH | curveH | any | Count | fastM | fastS | curveM | curveS | curveC |
| 0-0 | 0.00 | 0.30 | 0.70 | 0.00 | 0.00 | 0-0 | 0.00 | 0.20 | 0.80 | 0.00 | 0.00 |
| 1-0 | 0.00 | 0.30 | 0.70 | 0.00 | 0.00 | 1-0 | 0.00 | 0.20 | 0.80 | 0.00 | 0.00 |
| 0-1 | 0.00 | 0.00 | 1.00 | 0.00 | 0.00 | 0-1 | 0.00 | 0.20 | 0.50 | 0.30 | 0.00 |
| 1-1 | 0.00 | 0.00 | 1.00 | 0.00 | 0.00 | 1-1 | 0.00 | 0.20 | 0.50 | 0.30 | 0.00 |
| 2-1 | 0.00 | 0.00 | 1.00 | 0.00 | 0.00 | 2-1 | 0.00 | 0.20 | 0.50 | 0.30 | 0.00 |
| 3-1 | 0.00 | 0.60 | 0.40 | 0.00 | 0.00 | 3-1 | 0.10 | 0.00 | 0.70 | 0.20 | 0.00 |
| 2-0 | 0.00 | 0.60 | 0.40 | 0.00 | 0.00 | 2-0 | 0.10 | 0.00 | 0.70 | 0.20 | 0.00 |
| 3-0 | 0.70 | 0.30 | 0.00 | 0.00 | 0.00 | 3-0 | 0.70 | 0.00 | 0.30 | 0.00 | 0.00 |
| 0-2 | 0.00 | 0.00 | 0.00 | 0.00 | 1.00 | 0-2 | 0.00 | 0.00 | 0.20 | 0.60 | 0.20 |
| 1-2 | 0.00 | 0.00 | 0.00 | 0.00 | 1.00 | 1-2 | 0.00 | 0.00 | 0.20 | 0.60 | 0.20 |
| 2-2 | 0.00 | 0.00 | 0.00 | 0.00 | 1.00 | 2-2 | 0.00 | 0.00 | 0.40 | 0.60 | 0.00 |
| 3-2 | 0.00 | 0.00 | 0.00 | 0.00 | 1.00 | 3-2 | 0.40 | 0.00 | 0.60 | 0.00 | 0.00 |

### Table 12: The strategy of Team 10

| | Batter's strategy | | | | | | Pitcher's strategy | | | | |
|---|---|---|---|---|---|---|---|---|---|---|---|
| Count | wait | fastM | fastH | curveH | any | Count | fastM | fastS | curveM | curveS | curveC |
| 0-0 | 0.00 | 0.40 | 0.00 | 0.60 | 0.00 | 0-0 | 0.30 | 0.70 | 0.00 | 0.00 | 0.00 |
| 1-0 | 0.00 | 0.40 | 0.00 | 0.60 | 0.00 | 1-0 | 0.30 | 0.70 | 0.00 | 0.00 | 0.00 |
| 0-1 | 0.00 | 0.00 | 0.00 | 1.00 | 0.00 | 0-1 | 0.00 | 0.50 | 0.00 | 0.20 | 0.30 |
| 1-1 | 0.00 | 0.00 | 0.00 | 1.00 | 0.00 | 1-1 | 0.00 | 0.50 | 0.00 | 0.20 | 0.30 |
| 2-1 | 0.00 | 0.00 | 0.00 | 1.00 | 0.00 | 2-1 | 0.00 | 0.50 | 0.00 | 0.20 | 0.30 |
| 3-1 | 0.70 | 0.00 | 0.00 | 0.30 | 0.00 | 3-1 | 0.40 | 0.60 | 0.00 | 0.00 | 0.00 |
| 2-0 | 0.70 | 0.00 | 0.00 | 0.30 | 0.00 | 2-0 | 0.40 | 0.60 | 0.00 | 0.00 | 0.00 |
| 3-0 | 0.90 | 0.10 | 0.00 | 0.00 | 0.00 | 3-0 | 0.70 | 0.30 | 0.00 | 0.00 | 0.00 |
| 0-2 | 0.00 | 0.00 | 0.00 | 0.30 | 0.70 | 0-2 | 0.00 | 0.00 | 0.00 | 0.20 | 0.80 |
| 1-2 | 0.00 | 0.00 | 0.00 | 0.30 | 0.70 | 1-2 | 0.00 | 0.00 | 0.00 | 0.20 | 0.80 |
| 2-2 | 0.00 | 0.00 | 0.00 | 0.00 | 1.00 | 2-2 | 0.20 | 0.20 | 0.00 | 0.40 | 0.20 |
| 3-2 | 0.00 | 0.20 | 0.00 | 0.00 | 0.80 | 3-2 | 0.40 | 0.00 | 0.60 | 0.00 | 0.00 |

### Table 13: The strategy of Team 11

| | Batter's strategy | | | | | | Pitcher's strategy | | | | |
|---|---|---|---|---|---|---|---|---|---|---|---|
| Count | wait | fastM | fastH | curveH | any | Count | fastM | fastS | curveM | curveS | curveC |
| 0-0 | 0.00 | 0.40 | 0.00 | 0.60 | 0.00 | 0-0 | 0.00 | 0.20 | 0.80 | 0.00 | 0.00 |
| 1-0 | 0.00 | 0.40 | 0.00 | 0.60 | 0.00 | 1-0 | 0.00 | 0.20 | 0.80 | 0.00 | 0.00 |
| 0-1 | 0.00 | 0.00 | 0.00 | 1.00 | 0.00 | 0-1 | 0.00 | 0.20 | 0.50 | 0.30 | 0.00 |
| 1-1 | 0.00 | 0.00 | 0.00 | 1.00 | 0.00 | 1-1 | 0.00 | 0.20 | 0.50 | 0.30 | 0.00 |
| 2-1 | 0.00 | 0.00 | 0.00 | 1.00 | 0.00 | 2-1 | 0.00 | 0.20 | 0.50 | 0.30 | 0.00 |
| 3-1 | 0.70 | 0.00 | 0.00 | 0.30 | 0.00 | 3-1 | 0.10 | 0.00 | 0.70 | 0.20 | 0.00 |
| 2-0 | 0.70 | 0.00 | 0.00 | 0.30 | 0.00 | 2-0 | 0.10 | 0.00 | 0.70 | 0.20 | 0.00 |
| 3-0 | 0.90 | 0.10 | 0.00 | 0.00 | 0.00 | 3-0 | 0.70 | 0.00 | 0.30 | 0.00 | 0.00 |
| 0-2 | 0.00 | 0.00 | 0.00 | 0.30 | 0.70 | 0-2 | 0.00 | 0.00 | 0.20 | 0.60 | 0.20 |
| 1-2 | 0.00 | 0.00 | 0.00 | 0.30 | 0.70 | 1-2 | 0.00 | 0.00 | 0.20 | 0.60 | 0.20 |
| 2-2 | 0.00 | 0.00 | 0.00 | 0.00 | 1.00 | 2-2 | 0.00 | 0.00 | 0.40 | 0.60 | 0.00 |
| 3-2 | 0.00 | 0.20 | 0.00 | 0.00 | 0.80 | 3-2 | 0.40 | 0.00 | 0.60 | 0.00 | 0.00 |

## C    POSSIBLE BATTING RESULTS

We collect every piece of data of batters vs. pitchers from MLB data on Stacast Search in 2016. Then we classify all the possible batting results into 32 categories, as listed in Table 14, according to the batted ball information of "Event", "Location", "Type", "Distance", and "Exit Velocity" from collected data.

Batted Ball Type describes a batted ball, depending on how it comes off the bat and where in the field it lands. There are generally four descriptive categories: fly ball, pop-up, line drive, ground ball. Line drive is that a batted ball hit hard enough and low enough that it appears to travel in a relatively straight line. Pop-up is a specific type of fly ball that goes very high while not traveling very far laterally.

Table 14: The 32 categories of possible batting results

| category | description | location |
|---|---|---|
| **Hit** | | |
| 0 | HR | irrelevant |
| 1 | 3B | irrelevant |
| 2 | 2B | hit into a gap, far |
| 3 | 2B | down the foul lines |
| 4 | 2B | hit into a gap, not far |
| 5 | 1B | Left field |
| 6 | 1B | Center field |
| 7 | 1B | Right field |
| 8 | 1B | Infield |
| **Out** | | |
| 9 | ground ball | Pitcher |
| 10 | ground ball | Catcher |
| 11 | ground ball | First baseman |
| 12 | ground ball | Second baseman |
| 13 | ground ball | Third baseman |
| 14 | ground ball | Shortstop |
| 15 | line drive | Pitcher |
| 16 | line drive | Catcher |
| 17 | line drive | First baseman |
| 18 | line drive | Second baseman |
| 19 | line drive | Third baseman |
| 20 | line drive | Shortstop |
| 21 | fly ball | Left field |
| 22 | fly ball | Center field |
| 23 | fly ball | Right field |
| 24 | far fly ball | Left field |
| 25 | far fly ball | Center field |
| 26 | far fly ball | Right field |
| 27 | pop-up | irrelevant |
| **do not hit into play** | | |
| 28 | foul ball | --- |
| 29 | swing and a miss | --- |
| 30 | called strike | --- |
| 31 | ball | --- |

## D   SIMPLE BASEBALL RULE

In a half-inning, the batting team scores a "run" if a player legally runs around the bases and touches home plate. On the other hand, the fielding(defense) team tries to prevent runs by getting batters or runners "out". As soon as the fielding(defense) team gets three outs, the half-inning ends. There are usually nine innings in a baseball game, and two opposing teams take turn batting and fielding (defense). The team with the larger numbers of runs by the end of the game is the winner. The game proceeds when the pitcher in the fielding(defense) team throws the ball towards home plate and the batter decide whether to swing. A "strike" results when a batter swings and misses a pitch, does not swing at a pitch in the strike zone or hits a foul ball that is not caught. A "ball" results when a pitch misses the strike zone and the batter do not swing. A batter is out when getting three strikes, while is allowed to "walk" to the first base when getting four balls. There are lots of outcomes when the batter hit a ball into the field. In general, the batter aims to make solid contact to the pitch, while the pitcher aims to prevent the solid contact.

