# OpenReview forum: "Continuous Adaptation in Multi-agent Competitive Environments"
_ICLR.cc/2020/Conference — Reject_

### Official Review · AnonReviewer3 · 2019-10-22
**Official Blind Review #3**

**Rating:** 1

**Review:**

UPDATE: Thank you for the detailed response. I think the changes you have made to the paper have improved it, but there remains significant work to complete before the paper reaches its full potential. For example, Figure 3 is a useful additional insight but it provides no quantification of the variation between repeated runs nor a comparison to a suitable baseline approach. Similarly, the additional hyperparameter details in Appendix A are helpful to enable reproducability but without explaining how these values were chosen it is impossible to assess the rigor of the empirical evaluation.

--
This paper proposes modification to Deep CFR and introduce a simplified baseball environment to evaluate the modifications. The rigour of and detail in documenting the empirical evaluation is not currently of the standard I would expect for publication at ICLR. I will detail suggestions for improvement below, but my most pressing concern for discussion in the rebuttal is with regard to the 2nd conclusion - If the team has learnt a Nash equilibrium as initial strategy, then how can performance of either team be improved by only one team further adapting their policy as shown in Table 5 lower avg2 by the difference between No Adapt (0.61) and Guest Adapt (0.51)? If I have interpreted the authors results correctly, this demonstrates that the initial policy is not a Nash equilibrium.

Suggested Improvements:
In the related work section, the narrative of the paper would be clearer if the authors introduced why the meta-learning literature is being reviewed. I would also recommend weakening the claim that "instability of the training process" is "the critical issue in MARL" to just noting it is "an issue" and request the authors justify the claim that this "is particularly severe in a competitive environment". Why is this more of an issue in fully competitive environments than it is in general sum games?

In Section 3.1 the authors claim they need to switch to TD learning from tree search due to stochastic state transitions but there are forms of search that can accommodate stochastic transitions so this claim needs to be removed. Perhaps the authors can motivate this change in another way? This section also introduces 2 further simplifications of the domain (1) learning strategies for half-innings and applying them to the whole game; (2) agents knowing their opponents true action chosen and not the noisy observation (e.g. pitchers target location instead of actual pitched location) and (3) "both agents know each other's strategies"  - these should be included in Section 2.1 when the domain is described.

Section 3.1. closes by stating "In principle, their average strategies will gradually converge to the Nash-equilibrium strategy." This is very weakly argued, to make such a claim the authors should provide evidence that their environment and modifications to Deep CFR meet the requirements of the theory where this guarantee was proven.

Section 3.2. notes that the learning rate is "manually determined" but the precise methodology of tuning hyperparameters is not provided and no values of settings used for any hyperparameter are included in the paper. Without details of the methodolofy it is unclear if a rigorous empirical evaluation was performed and without the precise hyperparameter settings used the results are not reproducible.

Section 4.1. notes "It is very interesting to observe that the learned Nash-equilibrium strategy is actually quite similar to real-life baseball strategies." Setting aside the issue regarding whether a Nash equilibrium has been learnt, this is a subjective opinion not an rigorous empirical observation. The same applies to the comment "The simulation results are very similar to real-life baseball games" on page 9  Can you support these claims that the learnt strategy is similar to real-life by comparison to the data collected from the MLB Statcast?

In Section 4.2. there is further imprecision in the discussion of results. For example "as the strategy adaption mechanism is employed, the WP of most teams … actually decreases." As this mechanism is a core contribution of the paper, this evaluation needs to be more rigorous. Is there a statistically significant difference caused by using the proposed mechanism?

All results presented should include quantification of variation as well as average values and the number of repeats these averages are taken from should be clearly documented. The inclusion of a limited subset of teams in empirical evaluations (e.g. Table 4 only including teams 0,1,2,3 and 12 and Table 5 only including teams 5 and 13) should be clearly justified.

All references to papers published in conference proceedings or journals should cite the published version of the paper and not the arxiv version. All references should include the full publication venue and not abbreviations (e.g. Zhang and Lesser, 2010 is currently listed as just AAAI). References to online resources should include a note of the date accessed.

Minor Comments:
1) The word "purpose" is used frequently in place of "propose" (e.g. line 7 of the abstract)
2) Page 2: "plays the-best-of-three game" -> plays the best-of-three games
3) Page 2: "that worth further study" -> that are worth further study
4) Page 3: "To simply the game" -> To simplify the game
5) Page 3: "listed in C" -> listed in Appendix C
6) Page 8 (and recurring): Initial speech marks are the wrong way around, this looks like a Latex error.
7) Page 10: "we would relief" -> we would relax
8) Page 10: "in the real life" -> in real life


**Experience Assessment:**

I have published in this field for several years.

**Review Assessment: Checking Correctness Of Derivations And Theory:**

N/A

**Review Assessment: Checking Correctness Of Experiments:**

I carefully checked the experiments.

**Review Assessment: Thoroughness In Paper Reading:**

I read the paper thoroughly.

---

> ### Author Response · Authors · 2019-11-12
> **Response to Review #1 and Review #3**
>
> We would like to thank all the reviewers for offering their precious comments and suggestions. Listed below are the replies to these comments and brief descriptions of the major modifications in the revised manuscript.
>
> Question1: Why do we need adaptation if CFR already learns Nash?
> (Reviewer 1) Why do we need adaptation if CFR already learns Nash? It's unclear why the agents need to play something different than Nash. Could the authors argue why adaptation is necessary?
> (Reviewer 3) If the team has learnt a Nash equilibrium as initial strategy, then how can performance of either team be improved by only one team further adapting their policy as shown in Table 5 lower avg2 by the difference between No Adapt (0.61) and Guest Adapt (0.51)? If I have interpreted the authors results correctly, this demonstrates that the initial policy is not a Nash equilibrium.
> >>> Reply:
>     Both Nash equilibrium and opponent exploitation are fundamental problems in computational game theory. Nash equilibrium describes a static equilibrium strategy. However, in real games, players are often not under Nash equilibrium and we can do better via opponent exploitation by observing the weakness of the opponents. Take Rock–paper–scissors as an example. The Nash equilibrium is reached when BOTH players randomly choose rock, paper, and scissors with the probability 1/3 for each action. However, if one player has observed that his/her opponent has the tendency to play “rock” more often, he/she can adapt to play “paper” more frequently to increase the winning rate.
>     In this manuscript, we propose an adaptation mechanism for opponent exploitation based on a small number of observations in a multi-agent competitive environment. With the proposed adaptation mechanism, we further discuss the issue that “If two competitive teams adopt the same adaptation mechanism, what kind of initial strategy can help a team to quickly improve its winning percentage?” The experimental results show that if we pre-train the pitcher and the batter to follow a strategy that approximates the Nash equilibrium, then this team can quickly adapt its strategy to exploit various kinds of opponents based on the proposed adaptation mechanism. In our discussion, the opponents adopt other kinds of initial strategy.
>
> Question2: Do opponents actually play Nash?
> (Reviewer 1) Related to point 1, in section 4.2, it looks like post-adaptation strategies turn out to be superior when playing against opponents that play Nash. I would like to understand why. Do opponents actually play Nash? Does the asymmetry of the game have to do something with this?
> >>>Reply:
>     No, all the opponents do not play Nash. As explained in the reply to Question1, the main focus of our discussion is about the strategy adaptation mechanism for opponent exploitation, but not the static Nash equilibrium between two players.
>
> Comments:
> 1.	(Reviewer 1) I'm personally not familiar with baseball, and the paper doesn't really introduce the game. So, parts of the introduction and background that use baseball-specific terminology (paragraph 3) make no sense to readers unfamiliar with the game. It would be nice to have the game exemplified and explained along with the key simplifying assumptions.
> >>>Reply:
>     Many thanks for the suggestion. We have added a simple baseball rule in our manuscript in appendix D.
>
> 2.	(Reviewer 3) Section 3.1. closes by stating "In principle, their average strategies will gradually converge to the Nash-equilibrium strategy." This is very weakly argued, to make such a claim the authors should provide evidence that their environment and modifications to Deep CFR meet the requirements of the theory where this guarantee was proven.
> >>>Reply:
>     The CFR algorithm has been mathematically proven to converge to the Nash equilibrium if the training iteration comes close to the infinity. Unfortunately, baseball is a very complex game and so far we still haven’t provided a rigorous proof of this statement for the modified CFR algorithm. However, since our discussion focuses on the adaptation mechanism and the proper selection of the initial strategy, it is not our main purpose to ensure the learned strategy has eventually reached the Nash equilibrium. All we want to verify is that the strategy learned from the modified Deep CFR does serve as a good initial strategy in the proposed adaptation mechanism.
>     In the revised manuscript, we add in Figure 3 to demonstrate the improvement of the learned strategy in an empirical way. This figure shows that the winning percentage of the learned strategy against the other teams (Team 0 to Team 12) gets improved as iterations proceed. This implies that the learned strategy is getting closer to the Nash-equilibrium strategy if compared with the strategies of the other teams.

---

> ### Author Response · Authors · 2019-11-12
> **Response to Review #1 and Review #3**
>
> We would like to thank all the reviewers for offering their precious comments and suggestions. Listed below are the replies to these comments and brief descriptions of the major modifications in the revised manuscript.
>
> Comments:
> 3.	(Reviewer 3) In Section 4.2. there is further imprecision in the discussion of results. For example, "as the strategy adaption mechanism is employed, the WP of most teams … actually decreases." As this mechanism is a core contribution of the paper, this evaluation needs to be more rigorous. Is there a statistically significant difference caused by using the proposed mechanism?
> (Reviewer 1) There's virtually no analysis of the results in the paper, which significantly undermines any contribution.
> >>>Reply:
>     As aforementioned, we focus on the proper selection of the initial strategy and want to show that the strategy learned from the modified Deep CFR can serve as a good initial strategy in the proposed adaptation mechanism. In this manuscript, we only provide the phenomenon observed from our experiments. More detailed analyses of the adaptation mechanism will be investigated in the near future.
>
> 4.	(Reviewer 3) Section 3.1 introduces 2 further simplifications of the domain (1) learning strategies for half-innings and applying them to the whole game; (2) agents knowing their opponents true action chosen and not the noisy observation (e.g. pitchers target location instead of actual pitched location) and (3) "both agents know each other's strategies" - these should be included in Section 2.1 when the domain is described.
> >>>Reply:
>     In our original manuscript, Section 2.1 describes how we define our baseball game scenario, while Section 3.1 describes the “training method” that is used to learn the batter’s and pitcher’s strategies. Hence, we still prefer the division of these descriptions into two different sections.
>
> 5.	(Reviewer 3) Section 4.1. notes "It is very interesting to observe that the learned Nash-equilibrium strategy is actually quite similar to real-life baseball strategies." Setting aside the issue regarding whether a Nash equilibrium has been learnt, this is a subjective opinion not an rigorous empirical observation. The same applies to the comment "The simulation results are very similar to real-life baseball games" on page 9. Can you support these claims that the learnt strategy is similar to real-life by comparison to the data collected from the MLB Statcast?
> >>>Reply:
>     Many thanks for the reminding. This description is simply an interesting observation but not a claim. In the revised manuscript, we have revised the sentence to “It is very interesting to observe that the learned strategy at some states is quite similar to real-life baseball strategies.”
>
> 6.	(Reviewer 3) In the related work section, the narrative of the paper would be clearer if the authors introduced why the meta-learning literature is being reviewed.
> >>>Reply:
>     We have added a short paragraph at the beginning of Section 2.2 to introduce why meta-learning is important.
>
> 7.	(Reviewer 3) Section 3.2. notes that the learning rate is "manually determined" but the precise methodology of tuning hyperparameters is not provided and no values of settings used for any hyperparameter are included in the paper. Without details of the methodolofy it is unclear if a rigorous empirical evaluation was performed and without the precise hyperparameter settings used the results are not reproducible.
> >>>Reply:
>     Many thanks for the suggestion. We have added the setting of hyperparameters in Appendix A.

---

### Official Review · AnonReviewer1 · 2019-10-23
**Official Blind Review #1**

**Rating:** 1

**Review:**

The paper studies adaptation of agent policies in a simplified baseball game, which is designed as a zero-sum two-agent game between a batter (B) and a pitcher (P), each of which has 5 discreet actions. The introduced game is fully observable but stochastic, which the authors argue is a challenging setup. The authors propose a Bayesian-style adaptation of the agent strategies (where each agent models the probability of the actions of the opponent by computing the posterior give a prior and evidence from the past observations), which seems to be computable analytically, from an initialization learned with counterfactual regret minimization (CFR) that approximates the Nash of the considered game.

Comments/Questions:

1. Why do we need adaptation if CFR already learns Nash? It looks like one of the key differences between the proposed game/setup and many of the previous work is that it is a fully observable zero-sum game. The initialization learned by CFR already might come close to the Nash equilibrium. It's unclear why the agents need to play something different than Nash. Could the authors argue (preferably, formally theoretically or at least quantitatively) why adaptation is necessary?

2. Related to point 1, in section 4.2, it looks like post-adaptation strategies turn out to be superior when playing against opponents that play Nash. I would like to understand why. Do opponents actually play Nash? Does the asymmetry of the game have to do something with this? There's virtually no analysis of the results in the paper, which significantly undermines any contribution.


Other comments:

1. I'm personally not familiar with baseball, and the paper doesn't really introduce the game. So, parts of the introduction and background that use baseball-specific terminology (paragraph 3) make no sense to readers unfamiliar with the game. It would be nice to have the game exemplified and explained along with the key simplifying assumptions.

2. Writing can be significantly improved (and compressed!). There are typos throughout. Some phrases from the paper which meaning is really hard to parse (for example, "To focus on the impact of strategy adaptation over the winning percentage <...>").

Also, the last paragraph of the intro that describes the organization of the paper is unnecessary for conference submissions (it just takes spaces and no one reads it because it's easy to scroll through 10 pages to get a sense of the organization).

**Experience Assessment:**

I have published one or two papers in this area.

**Review Assessment: Checking Correctness Of Derivations And Theory:**

I assessed the sensibility of the derivations and theory.

**Review Assessment: Checking Correctness Of Experiments:**

I carefully checked the experiments.

**Review Assessment: Thoroughness In Paper Reading:**

I read the paper at least twice and used my best judgement in assessing the paper.

---

> ### Author Response · Authors · 2019-11-12
> **Response to Review #1 and Review #3**
>
> We would like to thank all the reviewers for offering their precious comments and suggestions. Listed below are the replies to these comments and brief descriptions of the major modifications in the revised manuscript.
>
> Question1: Why do we need adaptation if CFR already learns Nash?
> (Reviewer 1) Why do we need adaptation if CFR already learns Nash? It's unclear why the agents need to play something different than Nash. Could the authors argue why adaptation is necessary?
> (Reviewer 3) If the team has learnt a Nash equilibrium as initial strategy, then how can performance of either team be improved by only one team further adapting their policy as shown in Table 5 lower avg2 by the difference between No Adapt (0.61) and Guest Adapt (0.51)? If I have interpreted the authors results correctly, this demonstrates that the initial policy is not a Nash equilibrium.
> >>> Reply:
>     Both Nash equilibrium and opponent exploitation are fundamental problems in computational game theory. Nash equilibrium describes a static equilibrium strategy. However, in real games, players are often not under Nash equilibrium and we can do better via opponent exploitation by observing the weakness of the opponents. Take Rock–paper–scissors as an example. The Nash equilibrium is reached when BOTH players randomly choose rock, paper, and scissors with the probability 1/3 for each action. However, if one player has observed that his/her opponent has the tendency to play “rock” more often, he/she can adapt to play “paper” more frequently to increase the winning rate.
>     In this manuscript, we propose an adaptation mechanism for opponent exploitation based on a small number of observations in a multi-agent competitive environment. With the proposed adaptation mechanism, we further discuss the issue that “If two competitive teams adopt the same adaptation mechanism, what kind of initial strategy can help a team to quickly improve its winning percentage?” The experimental results show that if we pre-train the pitcher and the batter to follow a strategy that approximates the Nash equilibrium, then this team can quickly adapt its strategy to exploit various kinds of opponents based on the proposed adaptation mechanism. In our discussion, the opponents adopt other kinds of initial strategy.
>
> Question2: Do opponents actually play Nash?
> (Reviewer 1) Related to point 1, in section 4.2, it looks like post-adaptation strategies turn out to be superior when playing against opponents that play Nash. I would like to understand why. Do opponents actually play Nash? Does the asymmetry of the game have to do something with this?
> >>>Reply:
>     No, all the opponents do not play Nash. As explained in the reply to Question1, the main focus of our discussion is about the strategy adaptation mechanism for opponent exploitation, but not the static Nash equilibrium between two players.
>
> Comments:
> 1.	(Reviewer 1) I'm personally not familiar with baseball, and the paper doesn't really introduce the game. So, parts of the introduction and background that use baseball-specific terminology (paragraph 3) make no sense to readers unfamiliar with the game. It would be nice to have the game exemplified and explained along with the key simplifying assumptions.
> >>>Reply:
>     Many thanks for the suggestion. We have added a simple baseball rule in our manuscript in appendix D.
>
> 2.	(Reviewer 3) Section 3.1. closes by stating "In principle, their average strategies will gradually converge to the Nash-equilibrium strategy." This is very weakly argued, to make such a claim the authors should provide evidence that their environment and modifications to Deep CFR meet the requirements of the theory where this guarantee was proven.
> >>>Reply:
>     The CFR algorithm has been mathematically proven to converge to the Nash equilibrium if the training iteration comes close to the infinity. Unfortunately, baseball is a very complex game and so far we still haven’t provided a rigorous proof of this statement for the modified CFR algorithm. However, since our discussion focuses on the adaptation mechanism and the proper selection of the initial strategy, it is not our main purpose to ensure the learned strategy has eventually reached the Nash equilibrium. All we want to verify is that the strategy learned from the modified Deep CFR does serve as a good initial strategy in the proposed adaptation mechanism.
>     In the revised manuscript, we add in Figure 3 to demonstrate the improvement of the learned strategy in an empirical way. This figure shows that the winning percentage of the learned strategy against the other teams (Team 0 to Team 12) gets improved as iterations proceed. This implies that the learned strategy is getting closer to the Nash-equilibrium strategy if compared with the strategies of the other teams.

---

> ### Author Response · Authors · 2019-11-12
> **Response to Review #1 and Review #3**
>
> We would like to thank all the reviewers for offering their precious comments and suggestions. Listed below are the replies to these comments and brief descriptions of the major modifications in the revised manuscript.
>
> Comments:
> 3.	(Reviewer 3) In Section 4.2. there is further imprecision in the discussion of results. For example, "as the strategy adaption mechanism is employed, the WP of most teams … actually decreases." As this mechanism is a core contribution of the paper, this evaluation needs to be more rigorous. Is there a statistically significant difference caused by using the proposed mechanism?
> (Reviewer 1) There's virtually no analysis of the results in the paper, which significantly undermines any contribution.
> >>>Reply:
>     As aforementioned, we focus on the proper selection of the initial strategy and want to show that the strategy learned from the modified Deep CFR can serve as a good initial strategy in the proposed adaptation mechanism. In this manuscript, we only provide the phenomenon observed from our experiments. More detailed analyses of the adaptation mechanism will be investigated in the near future.
>
> 4.	(Reviewer 3) Section 3.1 introduces 2 further simplifications of the domain (1) learning strategies for half-innings and applying them to the whole game; (2) agents knowing their opponents true action chosen and not the noisy observation (e.g. pitchers target location instead of actual pitched location) and (3) "both agents know each other's strategies" - these should be included in Section 2.1 when the domain is described.
> >>>Reply:
>     In our original manuscript, Section 2.1 describes how we define our baseball game scenario, while Section 3.1 describes the “training method” that is used to learn the batter’s and pitcher’s strategies. Hence, we still prefer the division of these descriptions into two different sections.
>
> 5.	(Reviewer 3) Section 4.1. notes "It is very interesting to observe that the learned Nash-equilibrium strategy is actually quite similar to real-life baseball strategies." Setting aside the issue regarding whether a Nash equilibrium has been learnt, this is a subjective opinion not an rigorous empirical observation. The same applies to the comment "The simulation results are very similar to real-life baseball games" on page 9. Can you support these claims that the learnt strategy is similar to real-life by comparison to the data collected from the MLB Statcast?
> >>>Reply:
>     Many thanks for the reminding. This description is simply an interesting observation but not a claim. In the revised manuscript, we have revised the sentence to “It is very interesting to observe that the learned strategy at some states is quite similar to real-life baseball strategies.”
>
> 6.	(Reviewer 3) In the related work section, the narrative of the paper would be clearer if the authors introduced why the meta-learning literature is being reviewed.
> >>>Reply:
>     We have added a short paragraph at the beginning of Section 2.2 to introduce why meta-learning is important.
>
> 7.	(Reviewer 3) Section 3.2. notes that the learning rate is "manually determined" but the precise methodology of tuning hyperparameters is not provided and no values of settings used for any hyperparameter are included in the paper. Without details of the methodolofy it is unclear if a rigorous empirical evaluation was performed and without the precise hyperparameter settings used the results are not reproducible.
> >>>Reply:
>     Many thanks for the suggestion. We have added the setting of hyperparameters in Appendix A.

---

### Decision · Program_Chairs · 2019-12-19

**Decision:**

Reject

**Comment:**

This paper studies whether adopting strategy adaptation mechanisms helps players improve their performance in zero-sum stochastic games (in this case baseball). Moreover they study two questions in particular, a) whether adaptation techniques are helpful when faced with a small number of iterations and 2) what’s the effect of different initial strategies when both teams adopt the same adaptation technique. Reviewers expressed concerns regarding the fact that the author’s adaptation techniques improve upon initial strategies, which seems to indicate that their initial strategies were not Nash (despite the use of CFR). In the lack of theory of why this seems to happen at the current setup (and whether indeed the initial strategies are Nash and why do the improve), stronger empirical evidence from more rigorous experiments seem somewhat necessary for recommending acceptance of this paper.